# **Global high-resolution forest disturbance type dataset**

Li Wang<sup>1</sup>, Shidong Liu<sup>1\*</sup>, Wanjuan Song<sup>1</sup>, Jie Zhang<sup>2, 3</sup>, Shengping Ding<sup>4</sup> <sup>1</sup>State Key Laboratory of Remote Sensing and Digital Earth, Aerospace Information Research Institute, Chinese Academy of Sciences, Beijing 100094, China;

<sup>2</sup>School of Land Science and Technology, China University of Geosciences (Beijing), Beijing 100083, China;
 <sup>3</sup>Department of Earth System Science, Ministry of Education Key Laboratory for Earth System Modeling, Institute for Global Change Studies, Tsinghua University, Beijing 100084, China;
 <sup>4</sup>Faculty of Science, University of Copenhagen. Copenhagen 1350, Denmark

Correspondence to: Shidong Liu (liusd@aircas.ac.cn)

- Abstract. Forests play a pivotal role in global carbon cycling and biodiversity conservation, yet they face increasing disturbances from both anthropogenic and natural drivers. This study presents the first high-resolution (30-m) global forest disturbance dataset (GFD) for 2000–2020, classifying 11 disturbance types by integrating Landsat-based Continuous Change Detection and Classification (CCDC) time-series analysis with spatial metrics and machine learning. A total of 57,000 expert-validated samples were used to train and validate a decision tree model, achieving an overall accuracy of 94.88%. The
- results reveal that forestry disturbance (43.79±0.31%), shifting cultivation (24.32±0.28%), and forest fires (11.45±0.05%) dominate global forest loss. There are regional differences in global forest disturbance, such as farmland expansion in South America and Africa, forest fires in northern regions, and shifting cultivation in tropical regions. Disturbed forests span 1,247.06±11.18Mha, accounting for 30.87% of the global forest area. Notably, 2.76% of global forests were newly established, primarily in China, India, and Brazil. Spatial consistency analysis with existing datasets (R<sup>2</sup>=0.93) confirms the
- reliability of the GFD product. The GFD dataset advances our understanding of forest dynamics and underscores the need for targeted conservation strategies in an era of escalating environmental change. The 30 m resolution GFD generated by this study is openly available at https://doi.org/10.6084/m9.figshare.28465178 (Liu et al., 2025a).

#### **1** Introduction

- Forests, the dominant component of terrestrial ecosystems and the most widespread vegetation type on land, play a 25 pivotal role in delivering critical ecosystem services, including climate regulation (Piao et al., 2020; Xu et al., 2022), biodiversity conservation (Betts et al., 2017), soil and water retention, carbon sequestration (Tong et al., 2020), and habitat provision (Oeser et al., 2021). However, in recent decades, forest ecosystems have faced escalating disturbances from both natural drivers (Leverkus et al., 2018; Yan et al., 2022; Mayer et al., 2024) (droughts, extreme rainfall, and wildfires exacerbated by climate anomalies) and anthropogenic activities (deforestation, shifting cultivation, cropland expansion, and
- urbanization) (Acil et al., 2025; Chowdhury et al., 2017; Rivera et al., 2023; Liu et al., 2025b). These disturbances have severely compromised forest composition, structure, and functionality, thereby degrading their ecological services (Yang et

al., 2020; Feng et al., 2021). Consequently, accurate, timely, and continuous monitoring of forest disturbances is imperative for effective forest management, climate change mitigation, and global carbon accounting.

Forest disturbance represents one of the most critical processes in ecosystem succession (Ross et al., 2021; Mason et al., 35 2019; Blaschke et al., 1992), essential for maintaining regional ecological equilibrium (Reza and Abdullah, 2011; Kittel et al., 2000). Forest dynamics encompass two opposing processes: disturbance (forest cover loss or structural degradation caused by natural or human factors) and gain (forest recovery through natural regeneration or afforestation). Rapid population growth and urbanization have intensified conflicts between natural resource exploitation and human activities (Jiang et al., 2021; Miatto et al., 2021). Thus, characterizing the spatiotemporal patterns of forest disturbance and gain is vital for 40 understanding forest dynamics, estimating carbon stocks, and elucidating global change mechanisms (Chen et al., 2023b; Cuni-Sanchez et al., 2021; Peng et al., 2023). Given this context, high-accuracy identification of disturbance types has

Traditional forest monitoring predominantly relies on field surveys, which suffer from subjectivity, low temporal resolution, and high labor costs, rendering them inadequate for large-scale applications (Scheeres et al., 2023; Finger et al.,

emerged as a key scientific challenge in global environmental governance and sustainable development.

- 2021). Satellite remote sensing has revolutionized this field by offering extensive spatial coverage, continuous temporal observations, and rich spectral information (Zhao et al., 2023; Skidmore et al., 2021). Early remote sensing approaches, such as bi-temporal image comparison (post-classification change detection or spectral differencing), were limited by their sensitivity to image registration accuracy and inability to capture gradual disturbances (Wang et al., 2021). Pixel-based methods (NDVI thresholding) could detect vegetation changes but failed to discriminate disturbance types (deforestation,
- fires, or shifting cultivation).

Recent advances in time-series analysis have significantly improved monitoring capabilities (Tollerud et al., 2023; Liu et al., 2024). For instance, the Continuous Change Detection and Classification (CCDC) algorithm decomposes Landsat time-series data into trend, seasonal, and noise components, enabling disturbance detection at 30-m resolution (Tollerud et al., 2023; Hwang et al., 2022). Nevertheless, these methods exhibit notable limitations: inadequate spectral-temporal feature integration, leading to high confusion errors between plantation rotation and shifting cultivation; and poor model generalizability, algorithms like CUSUM, developed for temperate forests, underperform in tropical regions due to cloud contamination and phenological variability (Aquino et al., 2022; Ygorra et al., 2021).

To address these challenges, we propose a machine learning framework that synergizes time-series features with spatial aggregation metrics, leveraging the nonlinear modeling strengths of ensemble algorithms. This study aims to produce the

60 first high-resolution (30-m) global map of 11 major forest disturbance types (Table 1) in 2000–2020 by integrating Landsat CCDC time-series and spatial predictors within Google Earth Engine (GEE). To account for regional heterogeneity in forest types, climate regimes, and disturbance drivers, we partitioned the globe into four subregions for model training. Our results will directly support the Paris Agreement's carbon accounting framework, provide subtype data for platforms like Global Forest Watch (GFW) and Hansen's global forest change dataset, and inform regional forest restoration strategies.

55

Table 1: Global forest disturbance classification framework

| Code | Disturbance<br>type                    | Disturbance<br>intensity | Disturbance<br>source            | Forest type        | Disturbance process                                                                                                   | Recovery<br>type    |
|------|----------------------------------------|--------------------------|----------------------------------|--------------------|-----------------------------------------------------------------------------------------------------------------------|---------------------|
| 0    | Undisturbed                            | Undisturbed              | -                                | Natural<br>forests | Undisturbed between 2000 and 2020.                                                                                    | -                   |
| 11   | Shifting cultivation                   | Strong                   | Human<br>disturbance             | Natural forests    | Residents randomly cut down forests on a<br>small scale and plant crops, then abandon<br>cultivation after 1-2 years. | Natural recovery    |
| 12   | Forestry<br>disturbance                | Strong                   | Human<br>disturbance             | Natural<br>forests | To obtain wood, natural forests were cut down, and later manual planted them.                                         | Manual reversion    |
| 13   | Plantation disturbance                 | Strong                   | Human<br>disturbance             | Plantation         | Regular logging and renewal of plantations.                                                                           | Manual reversion    |
| 14   | Deforestation<br>of natural<br>forests | Strong                   | Human<br>disturbance             | Natural forests    | To obtain wood, natural forests were cut down, and later natural recovery.                                            | Natural recovery    |
| 15   | Forest fire<br>disturbance             | Strong                   | Natural fire                     | All forests        | The destruction of forests by wildfires.                                                                              | Natural recovery    |
| 16 * | Drought                                | Weak                     | Natural climate                  | All forests        | Forest degradation caused by drought.                                                                                 | -                   |
| 17 * | Forest pests and diseases              | Weak                     | Natural<br>pests and<br>diseases | All forests        | Forest degradation caused by pests and diseases.                                                                      | -                   |
| 18   | Built-up area<br>expansion             | Strong                   | Human<br>disturbance             | All forests        | Expansion of built-up areas encroach on forests.                                                                      | No recovery         |
| 19   | Cropland occupation                    | Strong                   | Human<br>disturbance             | All forests        | Expansion of cropland encroach on forests.                                                                            | No recovery         |
| 20   | Flood<br>disaster                      | Strong                   | Natural<br>flood                 | All forests        | Flood disasters encroach on forests.                                                                                  | Natural<br>recovery |
| 21   | Oil palm                               | Strong                   | Human<br>disturbance             | All forests        | Expansion of oil palm plantations encroach<br>on forests                                                              | Manual reversion    |
| 22   | Newly added<br>forest                  | Negative                 | Human<br>disturbance             | Non forest         | Artificially planting forests on non-forest land.                                                                     | Manual planting     |

Note: \* indicates weak disturbance type. Due to the spatial overlap between weak and strong disturbance types, this study did not consider weak disturbances.

#### 2. Materials and methods

#### 2.1 Study workflow

We developed a novel classification algorithm using machine learning within the GEE platform that integrates Landsatbased CCDC time-series analysis with spatial characteristics of forest cover to classify main distinct forest disturbance types globally. The model training and validation incorporated 57,000 expertly labeled samples of forest disturbance, which were visually interpreted by trained remote sensing specialists specializing in forest monitoring. Utilizing multi-temporal Landsat data in 2000-2020 and ancillary datasets (Section 2.2.5), we constructed a comprehensive feature set comprising 18

- disturbance indicators (Table 2). These features were systematically derived from both temporal and spatial dimensions, including: Overall characteristics of forest disturbance (OC), pre-disturbance forest conditions (PDC), post-disturbance recovery patterns (PDP), disturbance potential metrics (DP), land use/cover features (LUC), spatial contextual attributes (SC). All feature variables were preprocessed in GEE and subsequently resampled to correspond with the 57,000 samples. The classifier was locally trained using Python3.9, with rigorous validation performed at sample locations. Our classification
- approach employed a decision tree-based machine learning algorithm (CRAT), with accuracy metrics quantitatively assessed using independent test samples (Fig. 1).

#### Table 2 Global Forest Disturbance Characteristics Indicator

| Indicator type | Forest disturbance characteristic indicators |                                             |                                         |  |  |  |  |
|----------------|----------------------------------------------|---------------------------------------------|-----------------------------------------|--|--|--|--|
| OC             | Disturbance frequency                        | Average disturbance period                  | Number of segments                      |  |  |  |  |
| PDC            | Linear intercept before disturbance          | Internal fluctuations before<br>disturbance | Interannual trend before<br>disturbance |  |  |  |  |
| PDP            | Linear intercept after disturbance           | Internal fluctuations after<br>disturbance  | Interannual trend after<br>disturbance  |  |  |  |  |
| DP             | Forest fire area                             | Plantation area                             | Intensity of population                 |  |  |  |  |
| LUC            | 2020 Land Use /Cover                         | Forest cover in 2000                        | Forest cover in 2020                    |  |  |  |  |
| SC             | Longitude                                    | Latitude                                    | Disturbance partition                   |  |  |  |  |