# Peer review of "Global high-resolution forest disturbance type dataset"

_Earth System Science Data, 2025_

## Author Comment (AC6)

Reviewer #1:

**Comment #1**

It is useful to have maps of world distribution of different forest disturbance types and the authors provide a higher-resolution data set. The results appear reliable and mark a significant contribution to the state of the world's forests. 13 situations are recognised (Table 1); of these, two 'weak disturbances' (drought, pests&diseases) are not considered, so 11 are mapped in Fig.5, including 'undisturbed' and 'newly added forest'. Excluding undisturbed and new leaves 9 types of disturbance, of which 7 are covered in Fig.3 and Table 4 (accuracy of flood and oil palm not being evaluated).

**Response #1**

*Thanks for your recognition and constructive suggestions, which make our manuscript stronger. In this version, we have further revised the manuscript and addressed all your concerns. Please see the detailed point-by-point responses below.*

**Comment #2**

My criticisms are essentially confined to details of presentation and wording. It might be good to have more information on how the types are defined and how time series permit recognition of e.g. recovered areas. (**Revised**)

**Response #2**

*Thanks for your constructive comments. We have supplemented the manuscript with further details regarding the definition of forest disturbance types (Page 2-3, Line 62-81). The identification of each type relies on time-series characteristics, including pre-disturbance conditions, the disturbance process, and post-disturbance recovery patterns.*

*"The global forest disturbance classification framework is established through a comprehensive synthesis of key disturbance characteristics, including disturbance intensity, disturbance source, forest types affected, disturbance processes, and recovery type. Based primarily on disturbance intensity, disturbances are categorized into negative disturbance (newly added forest, 22), positive strong disturbance, and positive weak disturbance. According to the differences in disturbance sources, such as human activities, natural wildfires, climatic factors, insect and disease outbreaks, and flooding, weak disturbances are further differentiated into drought-induced disturbances (16) and forest pest and disease disturbances (17). Similarly, strong disturbances are subdivided into forest fires (15), flood disasters (19),*

*and human-induced forest disturbances. Depending on post-disturbance recovery status and land use type, human-induced disturbances are further distinguished into built-up area expansion (18) and cropland occupation (19), where forests are not restored. Taking into account the forest type disturbed, human-induced disturbances are also classified into renewal plantation (13) and oil palm expansion (21), both of which involve manual reversion. Based on the presence of short-term agricultural activities during the disturbance process, natural recovery secondary forests are categorized into natural forest deforestation (14) and shifting cultivation (11). Meanwhile, natural forest areas that were logged and then actively restored by humans are identified as forestry replanting (12)."*

**Table 1: Global forest disturbance classification framework**

| Code | Disturbance type | Disturbance intensity | Disturbance source | Forest type | Disturbance process | Recovery type |
|---|---|---|---|---|---|---|
| 0 | Undisturbed | Undisturbed | - | Natural forests | Undisturbed between 2000 and 2020. | - |
| 11 | Shifting cultivation | Strong | Human disturbance | Natural forests | Residents randomly cut down forests on a small scale and plant crops, then abandon cultivation after 1-2 years. | Natural recovery |
| 12 | Forestry replanting | Strong | Human disturbance | Natural forests | To obtain wood, natural forests were cut down, and later manual planted them. | Manual reversion |
| 13 | Plantation disturbance | Strong | Human disturbance | Plantation | Regular logging and renewal of plantations. | Manual reversion |
| 14 | Deforestation of natural forests | Strong | Human disturbance | Natural forests | To obtain wood, natural forests were cut down, and later natural recovery. | Natural recovery |
| 15 | Forest fire disturbance | Strong | Natural fire | All forests | The destruction of forests by wildfires. | Natural recovery |
| 16 * | Drought | Weak | Natural climate | All forests | Forest degradation caused by drought. | - |
| 17 * | Forest pests and diseases | Weak | Natural pests and diseases | All forests | Forest degradation caused by pests and diseases. | - |
| 18 | Built-up area expansion | Strong | Human disturbance | All forests | Expansion of built-up areas encroach on forests. | No recovery |
| 19 | Cropland occupation | Strong | Human disturbance | All forests | Expansion of cropland encroach on forests. | No recovery |
| 20 | Flood disaster | Strong | Natural flood | All forests | Flood disasters encroach on forests. | Natural recovery |
| 21 | Oil palm | Strong | Human disturbance | All forests | Expansion of oil palm plantations encroach on forests. | Manual reversion |
| 22 | Newly added forest | Negative | Human disturbance | Non forest | Artificially planting forests on non-forest land. | Manual planting |

*Note: * indicates weak disturbance type. Due to the spatial overlap between weak and strong disturbance types, this study did not consider weak disturbances.*

*For the identification of recovery areas, the line segments fitted by CCDC provide trend information on forest changes over each time period. In particular, the trend information during the post-disturbance phase can effectively indicate whether forest recovery has occurred.*

*Utilizing multi-temporal Landsat data in 2000-2020 and ancillary datasets (Section 2.2.5), we constructed a comprehensive feature set comprising 18 disturbance indicators (Table 2). These features were systematically derived from both temporal and spatial dimensions, including: Overall characteristics of forest disturbance (OC), pre-disturbance forest conditions (PDC), post-disturbance recovery patterns (PDP), disturbance potential metrics (DP), land use/cover features (LUC), spatial contextual attributes (SC).*

**Table 2 Global Forest Disturbance Characteristics Indicator**

| Indicator type | Forest disturbance characteristic indicators | | |
| --- | --- | --- | --- |
| OC | Disturbance frequency | Average disturbance period | Number of segments |
| PDC | Linear intercept before disturbance | Internal fluctuations before disturbance | Interannual trend before disturbance |
| PDP | Linear intercept after disturbance | Internal fluctuations after disturbance | Interannual trend after disturbance |
| DP | Forest fire area | Plantation area | Intensity of population |
| LUC | 2020 Land Use /Cover | Forest cover in 2000 | Forest cover in 2020 |
| SC | Longitude | Latitude | Disturbance partition |

**Comment #3**

On line133 the treatment of 'vacant areas' is worrying: more information on this is needed, how big an area is affected? (**Revised**)

**Response #3**

*Thanks for your constructive comments. For the "vacant areas," we have provided additional clarification. These areas actually represent missing areas that were not covered by the existing CCDC dataset. We have replaced the term "vacant areas" with "missing areas".*

*The supplementary manuscript content is as follows (Page 7, Line 147-149):*

*"The dataset provides extensive coverage of global forest areas, but small number of missing areas occur along the edges of some images, accounting for approximately 6% of the total global forest area. For the missing areas in the dataset,……"*

**Minor Comments**

**Comment #1**

101 '… America, South …' comma missing (**Revised**)

**Response #1**

*Thanks for your suggestion. We have revised the expression here (Page 6, Line 114-115):*

*"....four major clusters: Africa, Southeast Asia and Australia, Central America and South America, and the Northern Forest Region."*

**Comment #2**

132 Insert space before 'in' (**Revised**)

**Response #2**

*Thanks for your suggestion. We have revised the expression here (Page 7, Line 147):*

*[ee.ImageCollection("GOOGLE/GLOBAL_CCDC/V1")] in GEE.*

**Comment #3**

140 'Considering ⋯' -this sentence is incomplete, it is just a clause introducing something that is missing. (**Revised**)

**Response #3**

*Thanks for your suggestion. We have revised the expression here (Page 8, Line 159-161):*

*"Generally, a high spatial consistency is typically observed between disturbance types such as forest fires and plantation expansion and global fire and plantation distribution."*

**Comment #4**

156 'Meanwhile ⋯' is an incomplete sentence – just a clause. I suggest replacing with 'Weak disturbances in forest cover are highly time-bound.' (**Revised**)

**Response #4**

*Thanks for your suggestion. We have revised the manuscript according to your suggestion (Page 9, Line 191-192):*

*"Meanwhile, weak disturbances in forest cover are highly time-bound."*

**Comment #5**

160 Delete 'are not considered' - duplication. (**Revised**)

**Response #5**

*Thanks for your suggestion. We have revised the manuscript according to your suggestion (Page 9, Line 194-195):*

*"Therefore, this study did not consider these two weak disturbance types of drought disturbance and pest disturbance."*

**Comment #6**

166-169 This sentence misuses punctuation (: and ; are repeated). Please re-write. (**Revised**)

**Response #6**

*Thanks for your suggestion. We have revised the expression here (Page 9, Line 201-204):*

*"The specific process consists of two steps: decision tree generation and pruning. During the*

*decision tree generation phase, a tree is constructed from the training dataset and is grown to its maximum possible size. Subsequently, pruning is performed using the validation dataset to select the optimal subtree, with the minimization of the loss function serving as the criterion for pruning."*

**Comment #7**

Fig.3 There is space to replace codes with brief versions of types – e.g. 'plantation'. **(Revised)**

**Response #7**

*Thanks for your suggestion. We have revised the Fig.3 according to your suggestion (Page 12, Figure 3):*

[Figure]

*"Figure 3 Confusion Matrix of Global Forest Disturbance Classification"*

**Comment #8**

Table 4 118 should be 18 **(Revised)**

**Response #8**

*Thanks for your suggestion. We have revised the expression here (Page 13, Table 4):*

**Table 4 Accuracy Evaluation of GFD Mapping Results**

| Type | User 's Accuracy | Uncertainty (±) | Producer's Accuracy | Uncertainty (±) | Overall Accuracy |
|------|------------------|-----------------|---------------------|-----------------|------------------|
| 11 | 84.03% | 0.87% | 84.56% | 0.86% | 94.88%± |
| 12 | 93.07% | 0.40% | 90.92% | 0.45% | 0.17% |
| 13 | 96.53% | 0.56% | 97.07% | 0.52% | |
| 14 | 74.33% | 1.85% | 85.01% | 1.62% | |
| 15 | 98.31% | 0.32% | 98.49% | 0.31% | |
| 18 | 97.41% | 0.29% | 98.49% | 0.23% | |
| 19 | 98.37% | 0.32% | 96.73% | 0.45% | |

**Comment #9**

254-260 There should be a space before ± (**Revised**)

**Response #9**

*Thanks for your suggestion. We have revised the manuscript according to your suggestion (Page 13, Line 277-284):*

*"The overall accuracy reaches 94.88% (±0.17%), indicating robust model performance at the aggregate level (Table 4). Forest fire disturbance (98.31% ±0.32% user's accuracy, 98.49% ±0.31% producer's accuracy) and cropland occupation (98.37% ±0.32%, 96.73% ±0.45%) demonstrate the highest classification reliability. Forestry replanting shows strong(93.07% ± 0.40%, 90.92% ±0.45%), while shifting cultivation achieves moderate performance and slightly more variable accuracy (84.03% ±0.87%, 84.56% ±0.86%). Deforestation of natural forests exhibits the lowest user's accuracy (74.33% ±1.85%), suggesting significant confusion with other disturbance types, despite its relatively higher producer's accuracy (85.01% ±1.62%). Built-up area expansion shows nominally high accuracy (97.41% ±0.29%). These results highlight the model's effectiveness for dominant disturbance types.."*

**Comment #10**

260 Not a sentence: 'both ···' implies ' ···and' (**Revised**)

**Response #10**

*Thanks for your suggestion. We have revised the expression here (Page 13, Line 284):*

*"These results highlight the model's effectiveness for dominant disturbance types."*

**Comment #11**

268    'Western Siberian Plain in North America'  ?? (**Revised**)

**Response #11**

*Thanks for your comments. We have revised the expression here (Page 13, Line 290-291):*

"The evergreen coniferous forest exhibits significant disturbance in the central Cordillera Mountains, southern Labrador Plateau, Eastern European Plain, and Western Siberian Plain."

**Comment #12**

Fig.4   As each small symbol represents an area (grid square?), the colours must represent density.   So ha per ⋯ ?   Up to 1500 ha, so per at least 39 x 39 km.   Please state resolution of this & Fig.5. (**Revised**)

**Response #12**

*Thanks for your suggestion. We have added the resolution to the legend in Figure 4 &5. The spatial resolution of Figure 4 is 5.5km. The maximum value of our statistical results is 1500ha, which is $15km^2$, less than half of a grid area.*

"Figure 4 Global Forest Disturbance Distribution Map in 5.5km resolution."

"Figure 5 Global Forest Disturbance Classification Map in 30 m resolution."

**Comment #13**

Fig.5   'Forestry replanting ' is inconsistent with text (lines 284, 288 etc.), other Figures (8 & 9) and Table 1 ('Forestry disturbance') and does not seem to be used elsewhere. Actually 'forestry disturbance' is an unfortunate term for just one type of forest disturbance – disturbance as a disturbance type.   Could it be replaced throughout by 'forestry replanting', 'recovered disturbance' or just   'replanted' ? (**Revised**)

**Response #13**

*Thanks for your suggestion. We have replaced all 'forestry disturbance' in the manuscript with 'forestry replanting'. (Page 3, Line 74-75; Page 8, Line 168-169; Page 14, Line 297-298):*

"Meanwhile, natural forest areas that were logged and then actively restored by humans are identified as forestry replanting (12)."

"……have been preliminarily identified through research: undisturbed (0), shifting cultivation disturbance (11), forestry replanting (12), plantation disturbance……"

"The main types of global forest disturbance are forestry replanting (43.79%), shifting

*cultivation (24.32%), and forest fires (11.45%) (Fig. 5)."*

[Figure]

*Figure 5 Global Forest Disturbance Classification Map in 30 m resolution.*

**Comment #14**

284-293 Presumably Mha should be M ha (**Revised**)

**Response #14**

*Thanks for your suggestion. Yes, we agree with your suggestion. However, based on the opinions of other reviewers, we have removed unnecessary statements here.*

**Comment #15**

Fig. 6 caption    Insert 'Note varying scales.' (**Revised**)

**Response #15**

*Thanks for your suggestion. We have added "note varying scales" to the legend of Figure 6 as per your suggestion (Page 15, Line 314-316).*

*"Figure 6: Global Typical Forest Disturbance Statistics. a. is the cropland occupation on forests; b. is the disturbance caused by forest fires; c. is the disturbance of shifting cultivation; d. is the disturbance of plantations (excluding oil palm). These results are presented on a grid of 1.5° × 2.5°, and note varying scales."*

**Comment #16**

Fisg.6 & 7 maps show density, so it is necessary to state the unit area and (as these are rectangular) its dimensions. (**Revised**)

**Response #16**

*Thanks for your suggestion. We have supplemented the unit area and its dimensions in Figures*

*6 and 7 as per your suggestion (Page 15, Figure 6; Page 16, Figure 7).*

*"Figure 6: Global Typical Forest Disturbance Statistics. a. is the cropland occupation on forests; b. is the disturbance caused by forest fires; c. is the disturbance of shifting cultivation; d. is the disturbance of plantations (excluding oil palm). These results are presented on a grid of 1.5° × 2.5°, and note varying scales."*

*"Figure 7 Global Forest Disturbance Characteristics. a is recovered forest area; b is unrecovered disturbed area; c is undisturbed forest area; d is newly added forest area. These results are presented on a grid of 1.5° × 2.5°, and note varying scales."*
* * *
**Comment #17**

Fig.7   What is the rationale of having red = most in a & b, but red= least in c and d?   (For me, a, c and d might be considered 'good'; b is 'bad'.).   Fig. 6 was consistent with red = most, so readers are going to be confused here. (**Revised**)

**Response #17**

*Thanks for your suggestion. We have standardized the legends for all subgraphs. All subgraphs are 'red=most' (Page 15, Figure 6; Page 16, Figure 7).*

[Figure]

*"Figure 6: Global Typical Forest Disturbance Statistics. a. is the cropland occupation on forests; b. is the disturbance caused by forest fires; c. is the disturbance of shifting cultivation; d. is the disturbance of plantations (excluding oil palm). These results are presented on a grid of 1.5° × 2.5°, and note varying scales."*

[Figure]

*"Figure 7 Global Forest Disturbance Characteristics. a is recovered forest area; b is unrecovered disturbed area; c is undisturbed forest area; d is newly added forest area. These results are presented on a grid of 1.5° × 2.5°, and note varying scales."*

**Comment #18**

328-330 This is misleading, based on the inclusion of 'all' in Fig.8b. That should be replotted excluding 'All'. Consistency over the 5 types is thus much less, and the big deviation for Forest fire requires comment. (**Revised**)

**Response #18**

*Thanks for your suggestion. We have revised Figure 8 and, taking into account the opinions of other reviewers, we have removed unnecessary Figure 8b.*

[Figure]

*"Figure 8 Overall spatial consistency comparison with CDGFL."*

**Comment #19**

Figs. 8a, and 9a-d: Note that all show highly skewed distributions of both x and y variables. Calculating regressions on logarithmic scales would reduce the influence of the few high values. It would, however , increase the leverage of the numerous small values: a choice has to be made based on the absolute error margins of small versus large values.   Perhaps both types of regression should be presented. (**Revised**)

**Response #19**

*Thanks for your suggestion. We strongly agree with the viewpoint. We have added logarithmic scale scatter plots in the Appendix B. In fact, the logarithmic scale fitting results are better, which also highlights the accuracy of our conclusion (Page 20-22, Line 391-409).*

*"Appendix B*

*We compared the logarithmic proportional characteristics of forest cover under the same drivers and disturbance types across different global regions. To highlight the consistency of a large number of smaller values between GFD and CDGFL, we performed logarithmic operations on all indicators. According to 200 grids covering a wide range of forest areas worldwide, the proportion of GFD in each grid has a high consistency with the proportion of CDGFL, with a consistency coefficient of 0.81 (R2=0.83) (Fig. B1).*

[Figure]

*Figure B1 Overall spatial consistency comparison with CDGFL under logarithmic scale.*

*Under logarithmic scale, all GFD categories also exhibit strong spatial consistency with the existing CDGFL dataset (Fig. 9). We quantified the four dominant disturbance types with the largest proportions: forestry replanting, shifting cultivation, forest fire, and deforestation of natural forests (Fig. B2). The comparative analysis reveals that these four major disturbance types display high spatial agreement with the existing low-resolution CDGFL dataset, with the following metrics: shifting cultivation ($R^2$=0.80), forestry replanting ($R^2$=0.77), forest fire ($R^2$=0.90), and*

*deforestation of natural forest (R²=0.80). The spatial consistency fitting of various disturbance types at logarithmic scale is higher, which further supports the main conclusion of section 3.4.*

[Figure]

*Figure B2 Spatial consistency under different forest disturbance types under logarithmic scale. a-d represent the spatial consistency of between the GFD and the CDGFL in shifting cultivation, forestry replanting, forest fire, and deforestation of natural forest, respectively."*

---

## Author Comment (AC7)

Reviewer #2:

**Comment #1**

The manuscript describes a 30-m global forest disturbance dataset (11 disturbance types) for the time of 2000 to 2020. Disturbance is derived from Landsat data applying the CCDC analysis. My comments focus primarily on the accuracy assessment and area estimation components of the work. A primary area of improvement of the manuscript would be to provide a clear articulation of the sampling design used to collect the data for the accuracy assessment and area estimates. Without a clear description of the sampling design and additional details, it is impossible to ascertain how the accuracy and area estimates were obtained.

**Response #1**

*Thanks for your constructive suggestions, which make our manuscript stronger. We have incorporated all of your suggestions. The primary objective of this study is to produce a global map of forest disturbance, which aligns closely with the title of our manuscript. Accordingly, in response to specific comment 4 (comment #10), we have removed the section on area estimation as it was not directly relevant to the main focus of the paper. Additionally, we have provided a detailed description of the sampling of sample points used for map production and validation. This addition enhances the clarity of the mapping methodology and improves the overall accuracy of the data. In this version, we have further revised the manuscript and addressed all your concerns. Please see the detailed point-by-point responses below.*

**Major Comments**

**Comment #1**

Additional details related to the sampling design(s) must be provided. It is unclear how specifically the sample of 57,000 30-m sample units were selected for the model training and validation (Lines 72-73). In Section 2.3 (Lines 153-154), the text states that "8 individuals were uniquely responsible for selecting 8 types, while an additional 4 individuals conducted secondary confirmation of the selected samples." This text seems to be referring to the process of labeling the sample units, not explaining how the sample units were selected. Did these individuals actually choose which sample units (30-m pixels) were in the sample? There is no mention of randomization in the protocol for selecting the sample, and no details presented of whether strata are present, even though later in the manuscript stratified estimation formulas for accuracy metrics are provided (equations 5 through 10). To compound the confusion, the

Figure 3 confusion error matrix has a sample size of nearly 17,000, but there is no mention in the text of how these sample units were selected. Is it a random subset of the 57,000 mentioned earlier? Or are these 17,000 sample units entirely independent of the training sample of 57,000? It is essential to describe the sampling design(s) used to select these units. (**Revised**)

**Response #1**

*Thanks for your constructive suggestions, which make our manuscript stronger. We have now included a detailed description of the sampling point selection process in the manuscript (Page 8-9, Line 173-182; Appendix A). The 57,000 sample points were selected through manual interpretation. Specifically, we first generated preliminary pre-labeled points automatically and globally with the support of existing single-type auxiliary datasets, such as forest fire map. These pre-labeled points, though not fully accurate, served as initial references. Our interpreters then visually identified and marked the final accurate sample points within the vicinity of these pre-labeled locations. This approach not only improves labeling efficiency but also ensures both the accuracy and randomness of the sample points.*

*"The selection of sample points was conducted in two steps: automated random generation and visual verification. We automatically generated imprecisely pre-labeled sample points through global simple random sampling, based on the existing auxiliary dataset of single-class disturbances, such forest fire, plantation, and cropland occupation, etc. Although these pre-labeled sample points are uncertainty, they provide our interpreters with rapid and spatially randomized regions for sample point selection. Subsequently, our interpreters performed visual verification and interpretation to mark accurate sample points within the vicinity of these pre-labeled locations. This approach significantly enhances both the efficiency and randomness of the sampling process.  Undisturbed sample points are randomly generated and validated as stable pixels based on forest regions outside the Hansen's global forest change dataset. The sample point are evenly distributed in the global forest disturbance area (Appendix A).*

*For the challenging distinction of 'shifting cultivation', its identification relied on detecting unique cyclical patterns in the time series. Interpreters were trained to confirm three key characteristics within the high-resolution historical imagery. (1) Clear cyclical boundaries: evidence of alternating phases of forest (fallow), clearing/burning (clearance), and crops (cultivation) on the same parcel of land over multiple years; (2) Short-cycle land cover change: a complete cycle typically lasts a few years, distinguishing it from permanent deforestation for agriculture; and (3) Small-scale and fragmented spatial patterns: shifting cultivation plots are usually small, irregularly shaped, and interspersed with patches of mature forest. Sample points were only designated as shifting cultivation if they met multiple of these criteria*

*simultaneously to ensure accuracy. For the forest weak disturbance types caused by drought disturbance (16) and pest disturbance (17), their sample point selection needs to refer to high-resolution long-term remote sensing images."*

*"Appendix A*

*The selection of sample points was primarily based on the time-series changes observed in Landsat images from 2000 to 2020, supplemented by historical high-resolution imagery from Google Earth. Through extensive analysis, eight types of forest disturbances were preliminarily identified: undisturbed (0), shifting cultivation disturbance (11), forestry replanting (12), plantation disturbance (13), deforestation of natural forests (14), forest fire disturbance (15), built-up area expansion (18), and cropland occupation (19). A total of 57,000 sample points representing these disturbance types were visually interpreted. These sample points are evenly distributed across global forest disturbance areas (Fig. A).*

[Figure]

*Figure A Overall spatial consistency comparison with CDGFL under logarithmic scale. "*

*For the 17,000 validation sample points, the total set of 57,000 accurately interpreted samples in this study was randomly divided into two groups at a ratio of 7:3. A total of 40,000 samples were used for model training, and the remaining 17,000 were reserved for validation of the model results. This procedure was described in our previous version. We have further supplemented the relevant content in the manuscript (Page 10, Line 208-210).*

*"During the training and validation process of the model, the sample points we selected were divided into a training set and a validation set according to a 7:3 pattern. 40000 sample points*

*are used for model training, and another 17000 sample points are used to validate the model training results."*

**Comment #2**

I have several concerns with the Figure 3 confusion matrix, which I will list as separate items as follows:

a) It seems very unlikely that there would be no errors associated with the undisturbed class (which is class 0). Out of 3476 cases, there was never a commission error or omission error of "undisturbed" – this class is perfectly mapped. It seems implausible that disturbed and undisturbed forest can be classified with 100% accuracy. **(Revised)**

**Response #2**

*Thanks for your comment. This result is attributed to the application of masking procedures. This approach was designed to guarantee the reliability of the training samples, thereby improving the accurate identification of disturbed forest areas. We have supplemented relevant content in the manuscript (Page 12, Line 271-273; Page 9, Line 180-181; Page 10, Line 218-219). To ensure sample accuracy, the visually interpreted samples for the undisturbed category were deliberately selected from pixels exhibiting long-term stability. These pixels were also subjected to masking. We utilized forest change datasets, such as the Hansen et al. dataset, to exclude pixels that had undergone changes. Furthermore, the same masking protocol was applied to our final mapping products. Consequently, it is expected that the undisturbed class demonstrates nearly 100% accuracy.*

*"Here, we restricted our analysis to areas that had undergone forest disturbance. The observed 100% accuracy for the undisturbed class is attributable to the masking procedure applied using Hansen's Global Forest Change dataset."*

*"Undisturbed sample points are randomly generated and validated as stable pixels based on forest regions outside the Hansen's global forest change dataset. "*

*"The final GFD map, except for the newly added forest, uses Hansen's global forest change dataset as a mask to remove pixels that have not undergone forest changes."*

**Comment #3**

b) The confusion matrix is presented in terms of sample counts, which is reasonable if the sampling design is simple random. Yet the authors present formulas for stratified sampling (equations 5-10). In particular, equation (5) indicates how the cell proportions should be

estimated for a stratified sample, but that formula was not apparently used in the analysis. The confusion matrix should be presented in terms of the estimated pij (cell proportions) when stratified sampling is used. This concern links to comment 1 because the manuscript does not include description of the sampling design. (**Revised**)

**Response #3**

*Thanks for your suggestion. As stated in the response to comment 1, we adopted a simple random sampling design. We appreciate your suggestion, which will greatly help improve our manuscript. We have modified the corresponding formula (Page 10-11, Line 230-244).*

Overall accuracy (OA) derived from the overall error matrix of 11 forest disturbance types:

$$OA = \frac{\sum_{j=1}^{11} n_{jj}}{N} \tag{2}$$

User's accuracy for Class $i$ ($U_i$) (the number of sample points mapped to class $i$ with reference to class $i$)

$$U_i = \frac{n_{ii}}{n_{i.}} \tag{3}$$

Producer's accuracy for class $j$ (the number of sample points with class $j$ mapped to reference class $j$)

$$P_j = \frac{n_{jj}}{n_{.j}} \tag{4}$$

**2.5.2 Estimating accuracy**

The sampling variability associated with the accuracy estimates should be quantified by reporting standard errors. The variance estimators are provided below, and taking the square root of the estimated variance results in the standard error of the estimator. For overall accuracy, the estimated variance is:

$$V(O) = \frac{OA(1 - OA)}{N - 1} \tag{5}$$

For user's accuracy of map class $i$, the estimated variance is

$$V(U_i) = \frac{U_i(1 - U_i)}{n_{i.} - 1} \tag{6}$$

For producer's accuracy of reference class $j$, the estimated variance is

$$V(P_j) = \frac{P_j(1 - P_j)}{n_{.j} - 1} \tag{7}$$

**Comment #4**

c) Row and column totals need to be added to Figure 3. (**Revised**)

**Response #4**

*Thanks for your suggestion. We have added the total of rows and columns in Figure 3 as per your suggestion (Page 12, Figure 3):*

[Figure]

*"Figure 3 Confusion Matrix of Global Forest Disturbance Classification"*

**Comment #5**

d) It is unclear what the vertical color bar on the right of the figure represents (range from 0 to 40,000). Please remove it or explain what it is. (**Revised**)

**Response #5**

*Thanks for your suggestion. We have removed the vertical color bar in Figure 3 as per your suggestion (Page 12, Figure 3):*

[Figure]

*"Figure 3 Confusion Matrix of Global Forest Disturbance Classification"*

**Comment #6**

e) I will identify this comment as purely an opinion, but I am skeptical that a disturbance product can achieve the high accuracies reported. Accurately mapping forest change is exceedingly difficult, so to achieve user's and producer's accuracies of over 95% for many of these disturbance types doesn't seem possible. Comment 2a is related to this same concern.

**(Revised)**

**Response #6**

*Thanks for your comment. We acknowledge that accurately mapping forest disturbance types remains highly challenging. We also note that for certain complex disturbance types, such as shifting cultivation and deforestation, our accuracy is even less than 80%. Higher accuracy was achieved for types such as plantations and fires, which benefits from significant advances in global single-type disturbance mapping efforts, including existing plantation and forest fire map, etc. In our identification process, these relatively accurate auxiliary datasets were incorporated as features in the machine learning model, thereby improving accuracy for these*

*categories. Furthermore, permanently changed areas—such as forests converted permanently to cropland or built-up areas—also showed higher accuracy, owing to the use of highly reliable land cover datasets from around 2020. These auxiliary datasets substantially supported the identification of certain disturbance types. We have now enhanced the manuscript with detailed descriptions of the sources and applications of these datasets.*

*In Study workflow section (Page 4, Line 89-92 and Table2):*

*"These features were systematically derived from both temporal and spatial dimensions, including: …… disturbance potential metrics (DP), land use/cover features (LUC)……"*

Table 2 Global Forest Disturbance Characteristics Indicator

| Indicator type | Forest disturbance characteristic indicators | | |
|---|---|---|---|
| … | … | … | … |
| DP | Forest fire area | Plantation area | Intensity of population |
| LUC | 2020 Land Use /Cover | Forest cover in 2000 | Forest cover in 2020 |
| … | … | … | … |

*In Land use/cover dataset section (Page 6, Line 120-129 and Table3):*

*"To assess the mapping accuracy of global forest change areas, this study incorporated multiple authoritative land cover and forest cover products as reference datasets, including: (1) ESA WorldCover 2020,…… identifying 2020 forest cover distribution, and classifying non-forest land cover types in 2020. The detailed information of these datasets is systematically documented in Table 3."*

Table 3: Source of Land Cover Dataset

| Dataset | Resolution | Dataset source and main purpose |
|---|---|---|
| ESA WorldCover 2020 | 10m | Used to assist in identifying disturbances in cropland, built-up areas, etc. https://worldcover2020.esa.int/ |
| … | … | … |

*In Ancillary datasets section (Page 8, Line 157-164):*

*"Forest disturbance has strong disturbance sources. Therefore, using existing disturbance source datasets to assist in identifying typical forest disturbance types can effectively improve the accuracy of mapping results. Generally, a high spatial consistency is typically observed between disturbance types such as forest fires and plantation expansion and global fire and plantation distribution. This study uses global fire distribution datasets, artificial plantation distribution datasets, oil palm datasets, and other auxiliary methods to identify forest disturbance types. Meanwhile, there is a high correlation between population distribution and*

*forest disturbance. This study collected a forest disturbance potential dataset from three aspects: population density, forest fire distribution, and spatial distribution of oil palms."*

*In Result section (Page 12, Line 267-268):*

*"Owing to the high accuracy of current DP and LUC datasets, such as forest fire maps, plantation maps, and land use/cover maps, the identification accuracy of their corresponding disturbance types is considerably improved."*
* * *
**Comment #7**

The accuracy estimates reported on page 12 and in Table 4 are also a cause for concern.

a) It is evident that the stratified formulas were not used to estimate producer's accuracy and overall accuracy. If the sampling design is stratified and the stratified formulas were not used, these estimates would be incorrect. (**Revised**)

**Response #7**

*Thanks for your comment. Thank you very much for your comment. We have revised the formula expression in the paper and re-evaluated the accuracy estimates (Page 13, Line 276-286 and Table4):*

*"The accuracy assessment results reveal significant variations in classification performance across different forest disturbance types. The overall accuracy reaches 94.88% ($\pm$0.17%), indicating robust model performance at the aggregate level (Table 4). Forest fire disturbance (98.31% $\pm$0.32% user's accuracy, 98.49% $\pm$0.31% producer's accuracy) and cropland occupation (98.37% $\pm$0.32%, 96.73% $\pm$0.45%) demonstrate the highest classification reliability. Forestry replanting shows strong(93.07% $\pm$0.40%, 90.92% $\pm$0.45%), while shifting cultivation achieves moderate performance and slightly more variable accuracy (84.03% $\pm$0.87%, 84.56% $\pm$0.86%). Deforestation of natural forests exhibits the lowest user's accuracy (74.33% $\pm$1.85%), suggesting significant confusion with other disturbance types, despite its relatively higher producer's accuracy (85.01% $\pm$1.62%). Built-up area expansion shows nominally high accuracy (97.41% $\pm$0.29%). These results highlight the model's effectiveness for dominant disturbance types."*

*Table 4 Accuracy Evaluation of GFD Mapping Results*

| Type | User 's Accuracy | Uncertainty ($\pm$) | Producer's Accuracy | Uncertainty ($\pm$) | Overall Accuracy |
|---|---|---|---|---|---|
| 11 | 84.03% | 0.87% | 84.56% | 0.86% | 94.88%$\pm$ 0.17% |
| 12 | 93.07% | 0.40% | 90.92% | 0.45% | |

| | | | | |
|---|---|---|---|---|
| 13 | 96.53% | 0.56% | 97.07% | 0.52% |
| 14 | 74.33% | 1.85% | 85.01% | 1.62% |
| 15 | 98.31% | 0.32% | 98.49% | 0.31% |
| 18 | 97.41% | 0.29% | 98.49% | 0.23% |
| 19 | 98.37% | 0.32% | 96.73% | 0.45% |

**Comment #8**

b) It seems very likely that the standard error values are incorrect for several cases. For example, if we had a simple random sample with a sample size of n=17,000 (approximate sample size of matrix in Figure 3), the standard error of overall accuracy would be SQRT[(0.95)*(0.05)/17000]=0.0033 or 0.33%. The reported standard error for overall accuracy is 2.86% from line 253, nearly 10 times larger. The standard errors for producer's accuracy of Types 18 and 19 (approximately 20% and 15%) are suspiciously large given the large sample sizes for these two disturbance types. Lastly, the standard errors reported for user's accuracy also don't match what I calculate if I apply equation (7) to the data in Figure 3. Please re-check the standard error estimates to confirm. (**Revised**)

**Response #8**

*Thanks for your comment. Thank you very much for your comment. We have revised the formula expression in the paper and re-evaluated the accuracy estimates (Page 13, Line 276-286 and Table4). The latest results are as you calculated, and there is a significant improvement in the variance of various accuracy metrics. The standard error of overall accuracy is SQRT[(0.9488)* (1-0.9488)/16972]= 0.17%.*

*"The accuracy assessment results reveal significant variations in classification performance across different forest disturbance types. The overall accuracy reaches 94.88% (±0.17%), indicating robust model performance at the aggregate level (Table 4). Forest fire disturbance (98.31% ±0.32% user's accuracy, 98.49% ±0.31% producer's accuracy) and cropland occupation (98.37% ±0.32%, 96.73% ±0.45%) demonstrate the highest classification reliability. Forestry replanting shows strong(93.07% ±0.40%, 90.92% ±0.45%), while shifting cultivation achieves moderate performance and slightly more variable accuracy (84.03% ±0.87%, 84.56% ±0.86%). Deforestation of natural forests exhibits the lowest user's accuracy (74.33% ±1.85%), suggesting significant confusion with other disturbance types, despite its relatively higher producer's accuracy (85.01% ±1.62%). Built-up area expansion shows nominally high accuracy (97.41% ±0.29%). These results highlight the model's effectiveness*

*for dominant disturbance types.*"

**Comment #9**

c) Note that Type 18 in Table 4 is accidentally mis-labeled as "118" (**Revised**)

**Response #9**

*Thanks for your suggestion. We have revised the expression here (Page 13, Table 4):*

**Table 4 Accuracy Evaluation of GFD Mapping Results**

| Type | User 's Accuracy | Uncertainty (±) | Producer's Accuracy | Uncertainty (±) | Overall Accuracy |
|---|---|---|---|---|---|
| 11 | 84.03% | 0.87% | 84.56% | 0.86% | 94.88%± 0.17% |
| 12 | 93.07% | 0.40% | 90.92% | 0.45% | |
| 13 | 96.53% | 0.56% | 97.07% | 0.52% | |
| 14 | 74.33% | 1.85% | 85.01% | 1.62% | |
| 15 | 98.31% | 0.32% | 98.49% | 0.31% | |
| 18 | 97.41% | 0.29% | 98.49% | 0.23% | |
| 19 | 98.37% | 0.32% | 96.73% | 0.45% | |

**Comment #10**

Table 5 provides estimates of area of the GFD types. Presumably these are from the inadequately described "validation" sample. The Abstract should be revised to clarify what is presented in the manuscript. The manuscript's title suggests that the primary purpose of the manuscript is to present a new global forest disturbance dataset (i.e., a map). But key parts of the manuscript are sample-based estimates of area, which would use the disturbance map for stratification, but the key data are then the sample and disturbance type labels provided by the expert interpreters. For area estimation the role of the new disturbance map is secondary. If the main objective of the manuscript is to provide this global dataset, then sample-based area estimates would seem unnecessary and only the accuracy results would be necessary to present.

This same ambiguity is present in the Conclusion section. Lines 354-358 highlight the map of disturbance. But without any transition flagging the use of sample-based area estimation, Lines 358-360 then report sample-based estimates of area (Table 5) that use only the map through stratification of the sample. Please revise the Abstract and Conclusion to more clearly identify the purpose of the map and the role of sample-based area estimation to the objectives of the manuscript. (**Revised**)

**Response #10**

*Thank for your suggestion. We have removed unnecessary area estimation content and Table 5 as per your suggestion to highlight the focus of our research - the GFD map dataset. The abstract and conclusion also removed the $\pm$% area estimation section (Page 1, Line 15-19; Page 19, Line 362-364 ):*

*"The results reveal that forestry replanting (43.79), shifting cultivation (24.32%), and forest fires (11.45) dominate global forest loss. There are regional differences in global forest disturbance, such as farmland expansion in South America and Africa, forest fires in northern regions, and shifting cultivation in tropical regions. Disturbed forests span 1,247.06Mha, accounting for 30.87% of the global forest area. Notably, 2.76% of global forests were newly established, primarily in China, India, and Brazil."*

*"The results highlight forestry replanting (43.79%), shifting cultivation (24.32%), and forest fires (11.45%) as the dominant drivers of global forest cover changes, collectively accounting for nearly 80% of the total disturbed area. Meanwhile, the newly added forests worldwide account for 2.76% of the global forest disturbance area."*

**Technical Corrections**

**Comment #11**

1. Line 15: It is not clear whether the number to the right of the +/- is a standard error or a margin of error of a confidence interval. Please identify more clearly. (**Revised**)

**Response #11**

*Thanks for your comments. This is the confidence interval. Based on your previous suggestion (Comment #10), we have removed the $\pm$ percentage related to area estimation.*

**Comment #12**

2. Lines 19-20: The comparison to other datasets provides an evaluation of "agreement" or "consistency" with these other datasets. These other datasets are not "truth". Therefore,

agreement with these other datasets does not "confirm reliability" or convey "accuracy" but instead quantifies consistency with other datasets. (**Revised**)

**Response #12**

*Thanks for your suggestion. We have revised the expression here (Page 1, Line 19-20):*

*"The spatial consistency analysis ($R^2$ = 0.93) highlights a strong overall agreement between the GFD product and other datasets, while the GFD product offers superior spatial resolution."*

**Comment #13**

3. Line 43: What specifically is "subjective" about field surveys? The implication is that remote sensing is not subjective, but that would seem dubious because surely there are subjective components of remote sensing as well. (**Revised**)

**Response #13**

*Thanks for your suggestion. We have revised the expression here (Page 2, Line 44-46):*

*"Traditional forest monitoring predominantly relies on field surveys, which are limited by low temporal resolution and high labor costs, rendering them difficult to scale for large-area or frequent-assessment applications (Scheeres et al., 2023; Finger et al., 2021)."*

**Comment #14**

4. Lines 19, 225, 226, 321: This is a minor point, but stating that a comparison is made with "existing" datasets is not meaningful because we obviously cannot make a comparison to a dataset that does not exist. It would be better to use "other datasets" instead of "existing datasets". (**Revised**)

**Response #14**

*Thanks for your suggestion. We have addressed all similar issues in the manuscript (such as Page 11, Line 245-247):*

*"2.5.3 Comparison with other datasets*

*We compared GFD map with these other datasets to calculate the degree of agreement on typical forest disturbance types."*

**Comment #15**

5. Page 10, equation (10): This formula for the standard error of the estimated proportion of area does not match equation (10) presented in Olofsson et al. (2014). (**Revised**)

**Response #15**

*Thanks for your comment. Based on your previous suggestion (Comment #10), we have removed related content.*
* * *
**Comment #16**

6. Equation (11): The use of "UA" for the standard error will be confusing because it could easily be misread as an abbreviation for "User's Accuracy" and "UA" provides no obvious connection to standard error. (**Revised**)

**Response #16**

*Thanks for your comment. Based on your previous suggestion (Comment #10), we have removed related content.*
* * *
**Comment #17**

7. Equation (12): Please check this formula. It seems unlikely that there would be a "bar" above qi (indicating a mean) in the denominator but no "bar" above pi in that same denominator. (**Revised**)

**Response #17**

*Thanks for your suggestion. We have revised this formula according to your suggestion (Page 11, Line 260).*

$$R^2 = 1 - \frac{\sum_i^n (q_i - p_i)^2}{\sum_i^n (q_i - \bar{p}_i)^2} \tag{8}$$
* * *
**Comment #18**

8. Line 226: Because these other datasets are not "truth", comparisons to these datasets would represent "agreement" and "disagreement". Use of the term "errors" does not seem appropriate here. (**Revised**)

**Response #18**

*Thanks for your suggestion. We have revised the expression here (Page 11, Line 246-247):*

*"We compared GFD map with these other datasets to calculate the degree of agreement on typical forest disturbance types."*

**Comment #19**

9. Line 234: a space should be inserted between "s" and "p" in "asp". (**Revised**)

**Response #19**

*Thanks for your suggestion. We have revised the expression here (Page 11, Line 254-255):*

*"......each grid relative to the global forest loss area, denoted as $p_i$, where i represents the driver type......".*

**Comment #20**

10. Table 4: state what the +/- columns represent. (**Revised**)

**Response #20**

*Thanks for your suggestion. We have added corresponding explanations in Table 4 (Page13, Table4). The $\pm$ represent uncertainty of accuracy.*

*Table 4 Accuracy Evaluation of GFD Mapping Results*

| Type | User 's Accuracy | Uncertainty ($\pm$) | Producer's Accuracy | Uncertainty ($\pm$) | Overall Accuracy |
|------|------------------|---------------------|---------------------|---------------------|------------------|
| 11 | 84.03% | 0.87% | 84.56% | 0.86% | 94.88%± 0.17% |
| 12 | 93.07% | 0.40% | 90.92% | 0.45% | |
| 13 | 96.53% | 0.56% | 97.07% | 0.52% | |
| 14 | 74.33% | 1.85% | 85.01% | 1.62% | |
| 15 | 98.31% | 0.32% | 98.49% | 0.31% | |
| 18 | 97.41% | 0.29% | 98.49% | 0.23% | |
| 19 | 98.37% | 0.32% | 96.73% | 0.45% | |

**Comment #21**

11. Line 288: The meaning of "robust" precision is unclear. In what sense can precision be "robust"? (**Revised**)

**Response #21**

*Thanks for your comment. Based on your previous suggestion (Comment #10), we have removed this content related to area estimation.*

**Comment #22**

12. Line 326: "MEA" should be "MAE" and the word "only" should be removed from before "13%" as that is a value judgment of magnitude of the disagreement. (**Revised**)

**Response #22**

*Thanks for your suggestion. We have revised the expression here (Page 16, Line 334-335).*

*"From the perspective of error, the MAE and RMSE of the two are 13% and 19%, respectively (Fig. 8)."*

**Comment #23**

13. Panel b) of Figure 8 should be deleted or perhaps converted to a small table. The R^2, MAE, and RMSE values do not make much sense for only 6 data points and the "All Types" case must have a massive influence on the summary statistics. (**Revised**)

**Response #23**

*Thanks for your suggestion. We have removed Figure 8b as per your suggestion (Page 17, Figure 8).*

[Figure]

*Figure 8 Overall spatial consistency comparison with CDGFL.*

**Comment #24**

14. Lines 338-340: "MEA" should be "MAE" in multiple places. (**Revised**)

**Response #24**

*Thanks for your suggestion. We have revised the expression here (Page 17, Line 342-346).*

*"The comparative analysis reveals that these four major disturbance types display high spatial agreement with the existing low-resolution CDGFL dataset, with the following metrics: shifting cultivation ($R^2$=0.78, MAE=6.76%, RMSE=15.71%), forestry replanting ($R^2$=0.83, MAE=10.61%, RMSE=17.49%), forest fire ($R^2$=0.85, MAE=5.93%, RMSE=12.17%), and deforestation of natural forest ($R^2$=0.62, MAE=4.66%, RMSE=11.47%)."*

**Comment #25**

15. Throughout the manuscript the word "samples" is used incorrectly. The definition of "sample" in statistics is that it is a subset of n units selected from the population. The individual elements of that sample are "sample units", in this case a sample unit is a 30-m pixel. Thus, there are not 57,000 "samples" (e.g., Line 13), but one "sample" consisting of 57,000 sample units or sample pixels. This incorrect use of "samples" should be corrected throughout the manuscript. (**Revised**)

**Response #25**

*Thanks for your suggestion. We fully agree with your opinion. Due to the fact that our sample contains 57000 sample points. Therefore, we have replaced "samples" with "sample points" throughout the manuscript (such as Page 14, Line 86; Page 41, Line 176-179 ).*

*" The model training and validation incorporated 57,000 expertly labeled sample points of forest disturbance……"*
*"Although these pre-labeled sample points are uncertainty, they provide our interpreters with rapid and spatially randomized regions for sample point selection. Subsequently, our interpreters performed visual verification and interpretation to mark accurate sample points within the vicinity of these pre-labeled locations."*

---

## Author Comment (AC8)

Reviewer #3:

**General Comments:**

**Comment #1**

This manuscript integrated CCDC time series change detection method and CART model to map and identify forest disturbance type at a global scale. I have a few concerns on the validity and robustness of the proposed method.

**Response #1**

*Thanks for your recognition and constructive suggestions, which make our manuscript stronger. In this version, we have further revised the manuscript and addressed all your concerns. Please see the detailed point-by-point responses below.*

**Comment #2**

The detection of disturbed forest pixels solely depends on CCDC model. What's the accuracy of change detection? I wonder whether the change detection error and/or modelling uncertainty of CCDC will affect the subsequence disturbance type mapping? CCDC assumes NDVI of all the forest pixels can be quantified by a linear trend term and a harmonic seasonality term (Eq. 1). In fact, not all the pixels will perfectly fit into this assumed model, which would consequently affect the fitting performance of CCDC and therefore the subsequent disturbance mapping. (**Revised**)

**Response #2**

*Thanks for your constructive comments. We acknowledge the uncertainty of CCDC in change detection. To alleviate this issue, we did not solely rely on the feature indicators of CCDC before model training, but instead supplemented many other datasets such as forest fire distribution, land use types, forest changes, plantations, etc. This study has minimized the interference caused by CCDC fitting uncertainty as much as possible. We have added as much detailed information as possible in the manuscript (Page 7, Line 151-154; Page 4, Line 87-92; Table 2).*

*"In fact, not all the pixels will perfectly fit into this assumed model, which would consequently affect the fitting performance of CCDC and therefore the subsequent disturbance mapping. To address this issue, we also added feature indicators such as DP and LUC (Table 2) before machine learning training and classification to improve the robustness of the classification model."*

*"Utilizing multi-temporal Landsat data in 2000-2020 and ancillary datasets (Section 2.2.5), we constructed a comprehensive feature set comprising 18 disturbance indicators (Table 2). These features were systematically derived from both temporal and spatial dimensions, including: Overall characteristics of forest disturbance (OC), pre-disturbance forest conditions (PDC), post-disturbance recovery patterns (PDP), disturbance potential metrics (DP), land use/cover features (LUC), spatial contextual attributes (SC)."*

**Table 2 Global Forest Disturbance Characteristics Indicator**

| Indicator type | Forest disturbance characteristic indicators | | |
|---|---|---|---|
| OC | Disturbance frequency | Average disturbance period | Number of segments |
| PDC | Linear intercept before disturbance | Internal fluctuations before disturbance | Interannual trend before disturbance |
| PDP | Linear intercept after disturbance | Internal fluctuations after disturbance | Interannual trend after disturbance |
| DP | Forest fire area | Plantation area | Intensity of population |
| LUC | 2020 Land Use /Cover | Forest cover in 2000 | Forest cover in 2020 |
| SC | Longitude | Latitude | Disturbance partition |

**Comment #3**

Besides, in addition to CCDC, there are many change detection models available, such as BEAST, BFAST, and Landtrendr. Why did the author go with CCDC? Will applying different model end up with the same change detection outcomes? (**Explained**)

**Response #3**

*Thanks for your comments. All of these models are capable of detecting changes, and each algorithm has its specific advantages. Based on our work requirements, we need to be able to achieve rapid high-resolution forest change detection results on a global scale. It is important to note that these models generally exhibit very slow computational speeds when processing long-time-series, high-resolution remote sensing imagery at the global level. In contrast, a global fitted dataset based on the CCDC algorithm has already been generated, which significantly reduces both time and computational resources. Therefore, we selected the CCDC model. We believe that other models could also yield satisfactory results, however, the associated time costs are difficult to estimate.*

**Comment #4**

It seems that the authors only considered and mapped abrupt forest loss, while graduate forest changes (e.g., forest degradation) and forest gain (e.g., natural regrowth and afforestation) were only mapped. (**Revised**)

**Response #4**

*Thanks for your comments. This GFD map only includes abrupt forest disturbance types. For gradual weak disturbances, we propose these types in the forest disturbance classification framework, such as degradation caused by drought, pests and diseases, etc. This will be an independent topic for further research. We have added corresponding explanations in the manuscript (Page 9, Line 190-195).*

*"For the forest weak disturbance types caused by drought disturbance (16) and pest disturbance (17), their sample point selection needs to refer to high-resolution long-term remote sensing images. Meanwhile, weak disturbances in forest cover are highly time-bound. For example, the decline in vegetation index caused by a period of drought will quickly recover due to an increase in precipitation. At the global scale, it is currently limited by the availability of remote sensing images. We are unable to select relevant sample points through Landsat imagery. Therefore, this study did not consider these two weak disturbance types of drought disturbance and pest disturbance. This will be an independent topic for further research."*

**Comment #5**

Line 80: "CRAT" should be "CART" (**Revised**)

**Response #5**

*Thanks for your suggestion. We have revised the expression here (Page 13, Line 93-95):*

*"Our classification approach employed a decision tree-based machine learning algorithm (CART), with accuracy metrics quantitatively assessed using independent test sample points (Fig. 1)."*

**Comment #6**

5. Does the undisturbed area indicate no change has occurred in the pixel? What's the omission rate (or under-detection rate) of CCDC? (**Revised**)

**Response #6**

*Thanks for your comments. Undisturbed areas refer to pixels that have not undergone significant change, no abrupt forest loss or other major disturbances. The CCDC model's inherent uncertainty can also lead to a certain rate of omission errors. To mitigate this uncertainty, we incorporated multiple auxiliary datasets to reduce its impact, such as the Hansen et al. forest change dataset, among others. We have added corresponding explanations in the manuscript (Page 7, Line 151-154).*

*"In fact, not all the pixels will perfectly fit into this assumed model, which would consequently affect the fitting performance of CCDC and therefore the subsequent disturbance mapping. To*

*address this issue, we also added feature indicators such as DP and LUC (Table 2) before machine learning training and classification to improve the robustness of the classification model.”*

**Comment #7**

How does the proposed algorithm perform in Landsat images with dense and consistent cloud coverage (e.g., in tropical area)? (**Revised**)

**Response #7**

*Thanks for your comments. Our algorithm performs reasonably well in Landsat imagery with dense and consistent cloud cover, such as in tropical regions, though its efficacy is inherently dependent on image quality. In these areas, the predominant disturbance types, such as shifting cultivation and deforestation, exhibit lower identification accuracy compared to other disturbance categories, which is likely attributable to persistent cloud contamination. Nevertheless, our model still achieves an accuracy of nearly 80% or higher (Page 13, Line 179-183) in these challenging cloud conditions.*

*“Forestry replanting shows strong(93.07% ±0.40%, 90.92% ±0.45%), while shifting cultivation achieves moderate performance and slightly more variable accuracy (84.03% ±0.87%, 84.56% ±0.86%). Deforestation of natural forests exhibits the lowest user's accuracy (74.33% ±1.85%), suggesting significant confusion with other disturbance types, despite its relatively higher producer's accuracy (85.01% ±1.62%).”*

---

## Author Comment (AC9)

Reviewer #4:

**General Comments:**

**Comment #1**

General comment: The paper is generally well written and provide a useful dataset for the community with reasonable methods. I have only some minor comments below:

**Response #1**

*Thanks for your recognition and constructive suggestions, which make our manuscript stronger. In this version, we have further revised the manuscript and addressed all your concerns. Please see the detailed point-by-point responses below.*

**Minor Comments**

**Comment #1**

Line 127: "We collected Google Global Landsat based CCDC segments (1999-2019)." I don't understand this. I think CCDC segments were created by the authors. What do you mean by 'collected'? (**Revised**)

**Response #1**

*Thanks for your comment. The global fitting results of CCDC are a major feature of the GFD type in this study. Since CCDC-based fitted datasets have already been produced for most regions worldwide by previous studies, we utilized these existing data directly to reduce computational time and resource requirements, as repeated calculation was deemed unnecessary. For the missing areas in the dataset, we use the same CCDC fitting method to supplement them, to obtain the complete CCDC fitting results for the global forest area. Furthermore, it supports us in identifying the GFD types. We have supplemented corresponding details in the manuscript (Page 7, Line 141-151):*

*"We collected Google Global Landsat based CCDC segments (1999-2019). The dataset was created from the Landsat 5, 7, and 8 Collection-1, Tier-1, surface reflectance time series, using all daytime images between 1999-01-01 and 2019-12-31. Each image was preprocessed to mask pixels identified as cloud, shadow, or snow (according to the 'pixel_qa' band), saturated pixels, and pixels with an atmospheric opacity > 300 (as identified by the 'sr_atmos_opacity' and 'sr_aerosol' bands). We have removed duplicate pixels in the overlapping scenes between the north and south. The results were output in 2-degree tiles for all landmasses between -60 °*

*and +85 ° latitude. We can directly call this dataset [ee.ImageCollection("GOOGLE/GLOBAL_CCDC/V1")] in GEE. The dataset provides extensive coverage of global forest areas, but small number of missing areas occur along the edges of some images, accounting for approximately 6% of the total global forest area. For the missing areas in the dataset, the CCDC algorithm is used to complete them, thereby obtaining vegetation change characteristics covering all forest areas worldwide. Based on the segmented fitting results of these features, we extracted the OC, PDC, and PDP of each pixel separately (Fig. 2)."*

**Comment #2**

Line 131: deduplicated is very complex word. Try to rephrase. (**Revised**)

**Response #2**

*Thanks for your suggestion. We have revised the expression here (Page 7, Line 145-146):*

*"We have removed duplicate pixels in the overlapping scenes between the north and south. The results were output in 2-degree tiles for all landmasses between -60° and +85° latitude."*

**Comment #3**

Line 133: what do you mean by "vacant areas"? (**Revised**)

**Response #3**

*Thanks for your comment. It refers to the areas that are not covered in the CCDC dataset we collected, that is, the missing areas. We have revised the wording in the manuscript by replacing 'vacancy areas' with' missing areas' (Page 7, Line 147-150):*

*"The dataset provides extensive coverage of global forest areas, but small number of missing areas occur along the edges of some images, accounting for approximately 6% of the total global forest area. For the missing areas in the dataset, the CCDC algorithm is used to complete them, thereby obtaining vegetation change characteristics covering all forest areas worldwide."*

**Comment #4**

Line 160: suggest removing drought and pest from Table 1 to avoid potential confusions. Linked to line 181, there it says there are 11 disturbance types. If you remove drought and pest from Table 1, then there remains 10 types. More confusing is that Fig 3 contains 10 types including the Code 0. Could you clarifiy this? (**Explained and Revised**)

**Response #4**

*Thanks for your comment. Our 11 types of perturbations actually include undisturbed types with code 0. For forests, sustained and stable undisturbed forest areas should also be a key area of focus. Therefore, we also set it as a special GFD type. For disturbances such as drought and pests and diseases, although not included in our GFD map, they are indispensable in sorting out the main types of global forest disturbances. This is also to maintain the integrity of the forest disturbance framework. We have supplemented the process of organizing and developing the forest disturbance framework table in the manuscript (Page 2-3, Line 62-75).*

*"The global forest disturbance classification framework is established through a comprehensive synthesis of key disturbance characteristics, including disturbance intensity, disturbance source, forest types affected, disturbance processes, and recovery type. Based primarily on disturbance intensity, disturbances are categorized into negative disturbance (newly added forest, 22), positive strong disturbance, and positive weak disturbance. According to the differences in disturbance sources, such as human activities, natural wildfires, climatic factors, insect and disease outbreaks, and flooding, weak disturbances are further differentiated into drought-induced disturbances (16) and forest pest and disease disturbances (17). Similarly, strong disturbances are subdivided into forest fires (15), flood disasters (19), and human-induced forest disturbances. Depending on post-disturbance recovery status and land use type, human-induced disturbances are further distinguished into built-up area expansion (18) and cropland occupation (19), where forests are not restored. Taking into account the forest type disturbed, human-induced disturbances are also classified into renewal plantation (13) and oil palm expansion (21), both of which involve manual reversion. Based on the presence of short-term agricultural activities during the disturbance process, natural recovery secondary forests are categorized into natural forest deforestation (14) and shifting cultivation (11). Meanwhile, natural forest areas that were logged and then actively restored by humans are identified as forestry replanting (12)."*

*Meanwhile, we have clarified that the GFD map used in this study does not include any disturbances (Page 3, Table1; Page 9, Line 190-195).*

*Table 1: Global forest disturbance classification framework*

| Code | Disturbance type | Disturbance intensity | Disturbance source | Forest type | Disturbance process | Recovery type |
|------|------------------|----------------------|-------------------|-------------|--------------------|---------------|
| 0 | Undisturbed | Undisturbed | – | Natural forests | Undisturbed between 2000 and 2020. | – |
| 11 | Shifting cultivation | Strong | Human disturbance | Natural forests | Residents randomly cut down forests on a small scale and plant crops, then abandon cultivation after 1-2 years. | Natural recovery |
| 12 | Forestry replanting | Strong | Human disturbance | Natural forests | To obtain wood, natural forests were cut down, and later manual planted them. | Manual reversion |
| 13 | Plantation disturbance | Strong | Human disturbance | Plantation | Regular logging and renewal of plantations. | Manual reversion |

| 14 | Deforestation of natural forests | Strong | Human disturbance | Natural forests | To obtain wood, natural forests were cut down, and later natural recovery. | Natural recovery |
| 15 | Forest fire disturbance | Strong | Natural fire | All forests | The destruction of forests by wildfires. | Natural recovery |
| 16 * | Drought | Weak | Natural climate | All forests | Forest degradation caused by drought. | – |
| 17 * | Forest pests and diseases | Weak | Natural pests and diseases | All forests | Forest degradation caused by pests and diseases. | – |
| 18 | Built-up area expansion | Strong | Human disturbance | All forests | Expansion of built-up areas encroach on forests. | No recovery |
| 19 | Cropland occupation | Strong | Human disturbance | All forests | Expansion of cropland encroach on forests. | No recovery |
| 20 | Flood disaster | Strong | Natural flood | All forests | Flood disasters encroach on forests. | Natural recovery |
| 21 | Oil palm | Strong | Human disturbance | All forests | Expansion of oil palm plantations encroach on forests | Manual reversion |
| 22 | Newly added forest | Negative | Human disturbance | Non forest | Artificially planting forests on non-forest land. | Manual planting |

Note: * indicates weak disturbance type. Due to the spatial overlap between weak and strong disturbance types, this study did not consider weak disturbances.

"For the forest weak disturbance types caused by drought disturbance (16) and pest disturbance (17), their sample point selection needs to refer to high-resolution long-term remote sensing images. Meanwhile, weak disturbances in forest cover are highly time-bound. For example, the decline in vegetation index caused by a period of drought will quickly recover due to an increase in precipitation. At the global scale, it is currently limited by the availability of remote sensing images. We are unable to select relevant sample points through Landsat imagery. Therefore, this study did not consider these two weak disturbance types of drought disturbance and pest disturbance. This will be an independent topic for further research."

**Comment #5**

Line 177: 200 should be 2001? (**Revised**)

**Response #5**

Thanks for your suggestion. We have revised the expression here (Page 10, Line 213):

"……considering the dynamic changes of flood inundation areas from 2001 to 2020, the forest areas that have been submerged……"

**Comment #6**

Figure 1: Change the disturbance type code to its name ? (**Revised**)

**Response #6**

Thanks for your suggestion. We have revised the Fig.1 and 3 according to your suggestion (Page 5, Figure 1; Page 12, Figure 3):

[Figure]

*Figure 1 Study workflow*

[Figure]

*"Figure 3 Confusion Matrix of Global Forest Disturbance Classification"*
* * *
**Comment #7**

Figure 4: what is the spatial resolution of this map? Better to show forest loss and forest expansion independently. If both forest loss and gain occur in the same grid cell of the map, how did you do? The legend shows only the area being 'disturbed' but it does not show the direction of forest cover change. (**Revised**)

**Response #7**

*Thanks for your suggestion. The resolution of Figure 4 is 5.5km. The forest disturbance identified in this study is the recognition of the entire disturbance process, rather than simply detecting forest loss and gain. Except for permanent deforestation and encroachment, in fact, most types of forest disturbance involve two processes: disturbance and restoration. For example, natural forest deforestation includes both the logging process and the restoration process of secondary forests. If there is no restoration of secondary forests, it belongs to other disturbance types, such as Cropland encroachment, etc. Similarly, the disturbance of plantations includes both the logging of existing forests and the planting of artificial plantations. We have added resolution information in the legend of Figure 4.*

[Figure]

*"Figure 4 Global Forest Disturbance Distribution Map in 5.5km resolution."*

*In fact, we have summarized in Figure 7 where areas have recovered after forest disturbance (loss first, gain later), where areas have not recovered (loss), and where areas have added new forests (gain) (Figure 7).*

[Figure]

*"Figure 7 Global Forest Disturbance Characteristics. a is recovered forest area; b is unrecovered disturbed area; c is undisturbed forest area; d is newly added forest area. These results are presented on a grid of 1.5° × 2.5°, and note varying scales."*

**Comment #8**

Section titles of 2.4.1 and 2.4.2 can be improved because readers don't know what are 'other types' of forest disturbance in contrast to those been described in 2.4.1. In this sense, the section title of 2.4.1 can be also improved to enhance readability. (**Revised**)

**Response #8**

*Thanks for your suggestion. We have revised the expression here (Page 9, Line 197; Page 10,*

*2.4.1 CART-based classification of core forest disturbance types*

*2.4.2 Identification of supplementary forest disturbance types*

**Comment #9**

Section 2.3 describes how training samples are derived no? This should be made clear in its title. (**Revised**)

**Response #9**

*Thanks for your suggestion. We agree that the original title of Section 2.3 was not precise enough. We have now revised the title to "2.3 Derivation of training and validation sample points" to more clearly reflect the content of this section, which indeed describes the method for deriving the training samples.*

**Comment #10**

Could you show a map describing the spatial distribution of the training samples? (**Revised**)

**Response #10**

*Thanks for your suggestion. We have presented the map describing the spatial distribution of the training samples in Appendix A (Page 20).*

*"The selection of sample points was primarily based on the time-series changes observed in Landsat images from 2000 to 2020, supplemented by historical high-resolution imagery from Google Earth. Through extensive analysis, eight types of forest disturbances were preliminarily identified: undisturbed (0), shifting cultivation disturbance (11), forestry replanting (12), plantation disturbance (13), deforestation of natural forests (14), forest fire disturbance (15), built-up area expansion (18), and cropland occupation (19). A total of 57,000 sample points representing these disturbance types were visually interpreted. These sample points are evenly distributed across global forest disturbance areas (Fig. A)."*

[Figure]

*Figure A Overall spatial consistency comparison with CDGFL under logarithmic scale.*

**Comment #11**

How the samples of ꞌshifting cultivationꞌ are determined? This is critical because we know that this type is quite challenging. (**Revised**)

**Response #11**

*Thanks for your comments. We fully agree that accurately identifying shifting cultivation samples is among the most challenging tasks in remote sensing-based disturbance mapping. Our approach was designed specifically to address this complexity and ensure high sample purity. We have added a detailed selection process for migration agricultural samples in the manuscript (Page 9, Line 183-190):*

*"For the challenging distinction of 'shifting cultivation', its identification relied on detecting unique cyclical patterns in the time series. Interpreters were trained to confirm three key characteristics within the high-resolution historical imagery. (1) Clear cyclical boundaries: evidence of alternating phases of forest (fallow), clearing/burning (clearance), and crops (cultivation) on the same parcel of land over multiple years; (2) Short-cycle land cover change: a complete cycle typically lasts a few years, distinguishing it from permanent deforestation for agriculture; and (3) Small-scale and fragmented spatial patterns: shifting cultivation plots are usually small, irregularly shaped, and interspersed with patches of mature forest. Sample points were only designated as shifting cultivation if they met multiple of these criteria simultaneously to ensure accuracy."*

**Comment #12**

Fig. 6 & Fig. 7 should also show its spatial resolution. (**Revised**)

**Response #12**

*Thanks for your suggestion. We have added the resolution to the legend in Figure 6 &7. These*

*results are presented on a grid of 1.5° × 2.5° resolution.*

*"Figure 6: Global Typical Forest Disturbance Statistics. a. is the cropland occupation on forests; b. is the disturbance caused by forest fires; c. is the disturbance of shifting cultivation; d. is the disturbance of plantations (excluding oil palm). These results are presented on a grid of 1.5° × 2.5°, and note varying scales."*

*"Figure 7 Global Forest Disturbance Characteristics. a is recovered forest area; b is unrecovered disturbed area; c is undisturbed forest area; d is newly added forest area. These results are presented on a grid of 1.5° × 2.5°, and note varying scales."*

**Comment #13**

Fig 7: How do you determine the disturbed but not recovered forests? i.e., panel b, by using land cover map time series described in the Methods section? (**Explained and Revised**)

**Response #13**

*Thanks for your comment. Yes, we determined the vegetation change trend after forest disturbance based on CCDC fitting. This is categorized as post-disturbance recovery patterns (PDP) in methods section (Page 4, Line 89-97). The characteristic indicators of this type can be found in Table 2, which can provide detailed information on the recovery of forest disturbances for subsequent machine learning models.*

*"These features were systematically derived from both temporal and spatial dimensions, including: Overall characteristics of forest disturbance (OC), pre-disturbance forest conditions (PDC), post-disturbance recovery patterns (PDP), disturbance potential metrics (DP), land use/cover features (LUC), spatial contextual attributes (SC). All feature variables were pre-processed in GEE and subsequently resampled to correspond with the 57,000 sample points. The classifier was locally trained using Python3.9, with rigorous validation performed at sample point locations. Our classification approach employed a decision tree-based machine learning algorithm (CART), with accuracy metrics quantitatively assessed using independent test sample points (Fig. 1)."*

*Table 2 Global Forest Disturbance Characteristics Indicator*

| Indicator type | Forest disturbance characteristic indicators | | |
|---|---|---|---|
| OC | Disturbance frequency | Average disturbance period | Number of segments |
| PDC | Linear intercept before disturbance | Internal fluctuations before disturbance | Interannual trend before disturbance |
| PDP | Linear intercept after disturbance | Internal fluctuations after disturbance | Interannual trend after disturbance |
| … | … | … | … |

*Methodologically, forests that have been disturbed but not restored are mainly those whose*

*CCDC fitting line segments have not shown an upward trend after disturbance, indicating that vegetation restoration has not been detected. In terms of disturbance types, it mainly includes farmland occupation, built-up area expansion, etc. After forests are cut down, their land use types are directly and permanently changed. Specifically in Figure 7, we combine all types of disturbances that have not been restored after disturbance, resulting in a disturbed but unrecovered forest.*